# The Effect of Glycidyl Azide Polymer Grafted Tetrafunctional Isocyanate on Polytriazole Polyethylene Oxide-Tetrahydrofuran Elastomer and its Propellant Properties

**DOI:** 10.3390/polym12020278

**Published:** 2020-01-31

**Authors:** Jinghui Hu, Weiqiang Tang, Yonghui Li, Jiyu He, Xiaoyan Guo, Rongjie Yang

**Affiliations:** School of Materials Science and Engineering, Beijing Institute of Technology, Beijing 100081, China; 3120170524@bit.edu.cn (J.H.); tangweiqiangbit@163.com (W.T.); 18811786218@163.com (Y.L.); yrj@bit.edu.cn (R.Y.)

**Keywords:** polytriazole polyethylene oxide-tetrahydrofuran (PTPET), propellants, curing reagent, bonding

## Abstract

A new energetic curing reagent, Glycidyl azide polymer grafted tetrafunctional isocyanate (N100-*g*-GAP) was synthesized and characterized by FT-IR and GPC approaches. Polytriazole polyethylene oxide-tetrahydrofuran (PTPET) elastomer was prepared by N100-*g*-GAP and alkynyl terminated polyethylene oxide-tetrahydrofuran (ATPET). The resulting PTPET elastomer was fully characterized by TGA, DMA, FTIR and mechanical test. The above analysis indicates that PTPET elastomers using N100-*g*-GAP as curing reagent have the potential for use in propellants. The overall formulation test of the composite propellants shows that this curing system can effectively enhance mechanical strength and bring a significant improvement in the interface interaction between the RDX & AP particles and binder matrix.

## 1. Introduction

Glycidyl azide polymer (GAP) is generally used as the energetic binder in solid composite propellants with a variety of oxidizers, such as ADN, RDX, CL-20, and AP. N100 (tetrafunctional isocyanate) is a classic polyfunctional isocyanate curing reagent. The combination of GAP and N100 is a general cross-linked polyurethanes approach used in solid composite propellants [1]. The curing system between hydroxyl-terminated polyethylene oxide-tetrahydrofuran and kinds of isocyanates (e.g., toluene diisocyanate (TDI)) is the common approach to form a polyurethane matrix in propellants. However, this curing approach has an unavoidable drawback as it is inclined to react with moisture in the ambient environment that forms bubbles in the matrix by the evolution of gaseous [2]. Not only is partial isocyanate function consumed in vain, but bubbles in propellants also reduces combustion stability and energy density. Classic 1, 3-dipolar addition reactions (click chemistry) between azide and alkyne groups is an alternate method to overcome this disadvantage [3,4,5,6,7]. Recently, some researchers have already taken these routes to improve the properties of the composite propellants. Kristensen group has successfully used this approach to prepare isocyanate-free elastomer [8]. Deng’s teams have applied polytriazole curing system to solid composite propellant research [9]. Our team also has carried out a systematic study of the application of the click chemical curing system in solid composite propellants [10,11,12].

Using alkynyl terminated polyethylene oxide-tetrahydrofuran (ATPET) as a binder, GAP as a curing reagent is an effective method to solve that problem [12]. However, there brings an issue of poor utilization of azide groups in the study of the PTPET-GAP curing system that needs to be solved. It is well known that the properties of branched polymers are quite different from those of linear polymers [13]. The poor mechanical properties remain an issue in solid composite propellant of PTPET and linear GAP molecule. Since the previous GAPs are all linear molecules, it is expected that the structural morphology of the molecules will be improved by branching GAP. Grafting to form a branched polymer can increase the effective distribution of the azide group as compared to a linear polymer, in the hope that the efficiency of the azide group can be increased. As has been known to us, the mechanical properties of the solid composite propellants are mainly determined by the nature of the binder body and the interface between the solid filler and the binder matrix. Finding effective ways to improve the mechanical properties of composite propellants is the goal that scientists and engineers have been pursuing. At the same time, there is a weak interfacial binder between the binder matrix and the solid particle including AP and RDX. The interaction between the oxidizer particles and binder plays an important role in the propellant’s behavior of stress. To overcome the phenomenon of dewetting between the solid particles and the binder matrix, adding a suitable bonding reagent is an effective approach to improve the mechanical properties of the composite propellants [14,15,16,17]. Certain polar groups can effectively promote the bonding between the binder and the RDX particles [18]. Previous research has confirmed that carbamates in some polymer have an effect of bonding to oxidants [14,15]. Herein, the carbamates are the functional groups formed by the reaction between a hydroxyl and an isocyanate group.

Graft polymer refers to the product formed by chemical bonds on the macromolecular chain with appropriate branched or functional side groups. The combination of the properties from the main chain and the branch chain contributes to the overall performance. By grafting, two polymers with distinct properties can be bonded together to form a graft with special properties. Therefore, the graft modification of polymers has become a simple and effective method to extend the application field of polymers and improve the properties of polymer materials.

This study aims to design an azide curing reagent with a bonding effect that can be applied to the click chemistry, by combing the curing and bonding functions. By reaction between GAP and N100 to form a graft copolymer named N100-*g*-GAP, the purpose of finding a graft copolymer as a curing reagent can be achieved, and the strength of the elastomer cross-linked network might be enhanced. At the same time, the functional group between unreacted hydroxyl groups and ammonium perchlorate as well as RDX can form a hydrogen bond. This graft polymer can act as a bonding reagent to reduce the dehumidification behavior of solid particles.

## 2. Experimental

### 2.1. Materials

N100 (*Mn* = 743 g/mol, f = 4.05) and two kinds of GAP (*Mn* = 1000 g/mol and *Mn* = 4000 g/mol) were purchased from Liming Research & Design Institution of Chemistry (Luoyang, China). The alkynyl-terminated polyethylene oxide-tetrahydrofuran (ATPET, *Mn* = 4000 g/mol, f = 1.99) sample was received from the Beijing Institute of Technology Flame Retardant Technology Co., Ltd. (Beijing, China). Aluminum powders (Al), hexogen (RDX), A3 (BDNPA/BDNPF) and ammonium perchlorate (AP) were purchased from Xi’an North Hui’an Chemical Industries Co., Ltd. (Xi’an, China). All the above reagents were vacuum dried at 50 °C before use. Other reagents were purchased from Beijing Tongguang Reagent Co., Ltd. (Beijing, China) and used directly without further treatment.

### 2.2. Synthesis of N100-g-GAP

GAP (*Mn* = 1000 g/mol, 80 g, 0.08 mol) was placed in a three-necked flask with a mechanical stirrer (500 r/min), and the temperature was slowly raised to 80 °C. N100 (14.7 g, 0.0198 mol) was slowly added into the flask through a funnel. And three drops of trimethylamine (TEA) were added as a catalyst. The mixture was stirred at 90 °C for 21 h. The TEA was removed by vacuum distillation. An orange viscous liquid N100-*g*-GAP was obtained after vacuum drying at 50 °C in a yield of 96%.

### 2.3. Preparation of Polytriazole Polyethylene Oxide-Tetrahydrofuran Elastomer

The PTPET elastomers were obtained by curing N100-*g*-GAP and GAP (*Mn* = 4000 g/mol) with ATPET by the same functional group ratio of azide to alkyne. The PTPET was cured into an elastomer that was 15 cm long, 6 cm wide, and 4 mm thick at 50 °C for seven days. The synthetic route of the N100-*g*-GAP and PTPET elastomer is illustrated in Scheme 1. Due to the fact that the bulk grafting reaction of the polymer cannot achieve complete reaction, the N100-*g*-GAP in Scheme 1 is a schematic diagram, and the product is a mixture having multiple grafting states.

### 2.4. Application of N100-g-GAP in PTPET Composite Propellants

The prepared N100-*g*-GAP was further applied to the PTPET composite propellants (consisting of RDX/AP/Al/A3/ATPET/N100-*g*-GAP). The above propellants had 76 wt% total solid content (29 wt% AP, 18 wt% Al and 29 wt% RDX), A3 plasticizer 12 wt% and 12 wt% ATPET-GAP binder system. After kneading and casting, the propellants slurry was cured at 50 °C for 7 d. The propellant’s were cut into dumb-bells of pieces based on GB/T 528, and the experimental analysis was carried out using a universal tensile machine.

### 2.5. Measurements and Analysis

FT-IR spectra were recorded on a Nicolet AVATAR 6700 spectrometer (Thermo Fisher Scientific, Waltham, MA, USA). The molecular weight and distribution were tested by gel permeation chromatography (GPC, Waters-1515, Milford, MA, USA). The mechanical properties of the samples were measured on a tension test (MTS SYSTEMS, Shanghai, China). Dynamic mechanical measurements (DMA) experiments were proceeded using a DMA/SDTA 861(METTLER, Greifensee, Switzerland) equipped (−90~60 °C) with a cup fixture of 1 Hz and a heating constant rate of 2 °C/min. Thermogravimetry analysis (TGA) was obtained by a Netzsch 209 F1 (Al_2_O_3_ crucible, Germany) instrument with the following test condition: test atmosphere of N_2_, the scanning rate of 10 °C/min and the temperature range of 40~500 °C. The microscopic interface properties of the propellant’s samples were researched by scanning electron microscopy (SEM, HITACHI TM3000, Tokyo, Japan).

## 3. Results and Discussion

### 3.1. IR and GPC Analysis of N100-g-GAP

The GPC curves of N100-*g*-GAP change in reaction time were shown in Figure 1A. The main peak of N100 gradually declines with the increase of reaction time and disappears completely at 21 h. The above phenomenon shows that most of the N100 groups are involved in the reaction. The height of the main peak (*Mn* = 4180 g/mol, *t* = 26.2 min) of the reaction product N100-*g*-GAP increases with the rise of the reaction time, and the curves become stable at 21 h. Comparing the outflow time of curves shifts to the left with time means that the molecular weight of the product N100-*g*-GAP is gradually increasing. It is clear that the polymer dispersity index (PDI) of the product N100-*g*-GAP is gradually broadened. H-NMR data of N100-*g*-GAP are listed in Appendix A.

The IR curves of N100-*g*-GAP change in reaction time as shown in Figure 1B. The stretching vibration peak of the hydroxyl group in GAP (*Mn* = 1000 g/mol) is at 3641 cm^−1^. It can be seen from the dynamic curve that this peak gradually becomes smaller as the degree of reaction increases, indicating that the hydroxyl group in GAP is gradually consumed in the reaction with -NCO. The peak at 3336 cm^−1^ belongs to the NH symmetric stretch, which is increasing with time. The asymmetric stretching vibration absorption peak in -NCO is at 2270 cm^−1^ and the stretching vibration of urea bond in C = O in -NCO is at 1681 cm^−1^. That is the typical characteristic peak of -NCO and its intensity decreases significantly as time passes. The peak at 770 cm^−1^ is attributed to the bending vibration peak in the carbamate formed by the reaction. Peak groups at 1722, 1533, and 1310 cm^−1^ are the stretching vibrational peak of the first, second and third zones in the formed amide. Most -NCO groups react with -OH to form carbamates, which can form hydrogen bonds with -NCO, -OH, etc., and have bonding effects. The above data confirmed that the copolymer N100-*g*-GAP has been successfully synthesized.

### 3.2. Thermal Stability Analysis of PTPET Elastomer

It is necessary to understand the thermal property of the PTPET elastomer which is an important parameter to consider the possibility for applications. TG-DTG data of the PTPET elastomers are summarized in Figure 2. Both curves show two decomposition stages. There is a weak peak at about 250 °C for these two samples that were caused by the decomposition of azide groups. This peak in PTPET (N100-*g*-GAP) curve is weaker than that of PTPET (GAP) caused by grafting N100 which reduces the density of azide groups in the molecule. The *T_5%_* temperature of both curves is around 250 °C, but the *T_max_* temperature of PTPET (N100-*g*-GAP) is 10 °C lower than that of PTPET (GAP). However, this decomposition temperature has already allowed for practical use as it is over 400 °C.

### 3.3. Dynamical Mechanical Analysis of the PTPET Elastomer

The loss factors (*tan δ*) of the PTPET elastomers prepared by N100-*g*-GAP and GAP (*Mn* = 4000 g/mol), measured by the DMA, are shown in Figure 3. It can be seen that both elastomers exhibit loss factor peaks in the figure, which locates at −59.02 °C and −60.22 °C, respectively, corresponding to the glass transitions of the PTPET elastomer. The movement of the polymer chain is hindered and the glass transition temperature is increased. Obviously, PTPET (N100-*g*-GAP) shows a slightly higher *T_g_* than PTPET (GAP). This phenomenon is supposedly due to the dense cross-linked network formed in PTPET (N100-*g*-GAP), which hinders the movement of the segment and increases the glass transition temperature. The cross-linked density of the PTPET will be confirmed in the swelling analysis in the next section. There is also a peak at −39.68 °C of the PTPET (GAP) curve due to the suspension GAP chain portion of the cross-linked network. The large molecular weight of the linear GAP becomes a longer segment after it formed in PTPET elastomer, and it shows β relaxation in DMA analysis. While there is no obvious β glass transition peak in PTPET elastomer prepared by N100-*g*-GAP. The length of the GAP segment is shortly after it is grafted onto N100, which is not enough to highlight the relaxation peak of its own segment.

### 3.4. Swelling Analysis of PTPET Prepared Elastomer

Prepared PTPET samples by N100-*g*-GAP and GAP (*Mn* = 4000 g/mol) were swollen in toluene at 25 °C. The swelling characteristics of samples were calculated according to the equilibrium swelling method [11,19,20]. The volume swelling curves of PTPET prepared by N100-*g*-GAP and GAP are displayed in Figure 4, which are the apparent molecular weight (*M_c_*) and the volume swelling ratio (*q_v_*) curves, respectively. The calculated parameter results appear in Table 1. The prepared both PTPET samples can reach equilibrium at 540 min. However, investigation results revealed that there was an obvious difference in the swelling phenomenon. The PTEPT (N100-*g*-GAP) showed a lower *q_v_* than PTPET (GAP). The cross-linked linking average molecular weight of PTPET (N100-*g*-GAP) was about 3510, which was lower than PTPET (GAP) with about 7405. The above results indicate that the cross-linking network of PTPET prepared by N100-*g*-GAP was denser and that more cross-linked points were formed [21].

### 3.5. Tensile Behaviors of PTPET Elastomer

The previous study shows the addition of a cross-linked network in the polymer structure can improve its strength due to the increase of cross-linked density [22]. The tensile analysis is an effective approach to investigate the cross-linked effect on the PTPET elastomer. A comparison of prepared PTPET sample’s strain and stress with the same content of cross-linked reagents is shown in Figure 5A (curves listed in Appendix A). The results at 20 °C illustrated that the PTPET (N100-*g*-GAP) elastomer had a better ability in stress than PTPET (GAP) elastomer, while the strain performance was reduced. The tensile experiment analysis explained that N100-*g*-GAP had a better ability in generating a cross-linked network than GAP in prepared the PTPET elastomer. Moreover, calculated parameters data from swelling analysis confirmed tensile experiment results.

Hydrogen bonds between the molecular chains exhibit a strong bonding force at low temperatures [23]. In order to judge whether there might be hydrogen bonds in the above PEPET (N100-*g*-GAP) elastomers, IR analysis was further carried out. The IR result of the carbonyl group is shown in Figure 5B. The carbonyl group and the active hydrogen in the elastomer have a double-associated hydrogen bond carbonyl and a single associative hydrogen bond carbonyl [24,25]. As seen from the curve of PTPET (N100-*g*-GAP), it is clear that the free carbonyl locates at 1725 cm^−1^. While for the peak of double-associated hydrogen bond urea carbonyl and mono-association hydrogen bond urea carbonyl were found at 1640 cm^−1^ and 1685 cm^−1^, respectively.

### 3.6. Thermal Stability Analyses of Propellants

The TG-DTG curves of the propellants prepared by N100-*g*-GAP and GAP were presented in Figure 6A,B. It can be found that the initial decomposition temperature (*T_5%_*) of the propellants prepared by GAP was at 204.9 °C, slightly lower than that of the N100-*g*-GAP at 206.7 °C. There were two main max decomposition peaks (*T_max_* at 228.67 °C and 308.67 °C) in the propellants prepared by N100-*g*-GAP during the DTG curve from Figure 6B. The *T_max_* of the propellants prepared by N100-*g*-GAP was slightly lower than GAP (231.78 °C). The TG-DTG curves indicated that there were two loss-wt% stages for propellants (N100-*g*-GAP), and that the 84.13% loss-wt% was resulted from the main breakage of the polyether backbone, while the 3.48 wt% was caused by the carbamate and part of the hard segment molecular chain. For the propellants (GAP), there is only one loss-wt% which was 84.70% that mainly due to the decomposition of the polyether polytriazole chain. Meanwhile, N100-*g*-GAP can replace the GAP to decrease the residual carbon by 4 wt% in propellants.

### 3.7. Dynamical Mechanical Analyses of Propellants

As shown in Figure 7, the damping factor (*tan δ*) was plotted as a function of temperature for different propellant samples. Propellants made by PTPET (GAP) and PTPET (N100-*g*-GAP) samples had measured *T_g_* temperature around −60.35 °C and −58.91 °C, respectively. That result was consistent with the swelling analysis that the high cross-linked density causes an increase in *T_g_* temperature of samples. The half-peak width of the *tan δ* curve of the propellants (PTPET (N100-*g*-GAP)) is broad, indicating that the cross-linked density of this propellant is higher than the propellants (PTPET (GAP)). The molecular weight distribution of the formed (PTPET (N100-*g*-GAP)) elastomer is wider than PTPET (GAP) in the composite solid propellant. The results obtained by DMA analysis are consistent with the previous GPC results. The analysis of interfacial interaction between solid particles and the PTPET elastomer matrix can be evaluated through DMA. It is clear that the damping coefficient can reflect the elastic response of the material and the dissipating energy losses. Decreased damping is evidence of improved interactions between solid filler and binder [16,26]. The damping of propellants (PTPET (N100-*g*-GAP)) shows a noticeable decrease compared with the propellants (PTPET (GAP)), clearly manifesting that N100-*g*-GAP raises adhesion of the PTPET (N100-*g*-GAP) binder and solid filler.

### 3.8. Mechanical Properties of Propellants

The mechanical test is an effective approach to testify how incorporation of N100-*g*-GAP would influence the composite propellants. A four-component propellant formulation containing RDX, AP, Al, and binder was selected to test, and the results were listed on the following Figure 8 (curves listed in Appendix A). It can be readily inferred that the use of N100-*g*-GAP can significantly improve the tensile strength of the composite propellants at 20 °C. At the same time, it can confirm that the elongation at break loss of the propellants was small, while the elongation at breaking was somewhat lowered. The comprehensive analysis showed that the mechanical strength of the propellants prepared by N100-*g*-GAP was higher than GAP, indicating that the cross-linked density of the prepared elastomer was higher, which was consistent with the results of the above swelling and DMA analysis. The mechanical strength advantage indicating that this may be due to the formation of hydrogen bonds between the PTPET (N100-*g*-GAP) and the solid filler in the composite solid propellant.

### 3.9. SEM Results of Propellants

In order to further access how N100-*g*-GAP-affects the interface effect of solid particles in composite propellants, a scanning electron microscope (SEM) was utilized to study the different samples by microscopy. Figure 9 shows photos of the fracture surfaces of the above two kinds of propellants specimens containing N100-*g*-GAP (Figure 9A,B), and GAP (Figure 9C,D) after tensile testing. It is obviously displayed that propellants with GAP have a distinctly poor adhesion of RDX &AP particle and binder matrix. There are no apparent inhomogeneities in the distribution of the PTPET (N100-*g*-GAP) elastomer between RDX and AP. The amino group on the carbamate can form a hydrogen bond with the N atom on the RDX and AP particles and acts as a bonding agent. The interface between N100-*g*-GAP, GAP and aluminum powder is not a good result. The above shows that N100-*g*-GAP can act as a binding reagent for RDX and AP propellants.

## 4. Conclusions

A new curing reagent N100-*g*-GAP with a higher crosslink effect for the PTPET system was successfully synthesized in this paper. Meanwhile, the urethane group formed in N100-*g*-GAP can form a hydrogen bond and has a bonding effect. Compared with GAP analysis, the PTPET elastomer prepared by N100-*g*-GAP has a higher crosslink density and thermodynamic stability. The use of N100-*g*-GAP significantly enhances the tensile strength of composite propellants at 20 °C. DMA and SEM analysis of propellant samples indicates an improved binder-filler interfacial interaction in the propellant composites containing N100-*g*-GAP.

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
