# Peer review of "The Effect of Glycidyl Azide Polymer Grafted Tetrafunctional Isocyanate on Polytriazole Polyethylene Oxide-Tetrahydrofuran Elastomer and its Propellant Properties"

_polymers, 2020, doi:10.3390/polym12020278_

Round 1

Reviewer 1 Report

This manuscript submitted by Guo et al. reported the synthesis and characterization of new curing agent naming N100-g-GAP as well as its application in PTPET elastomers/propellants. The elastomers were characterized by TGA, DMA, FTIR and mechanical test. However, this manuscript was limited by characterizations and improper discussions. I suggest reconsidering it after major revisions. Here are more comments.

The language needs major revisions in grammar. The description and discussion of experiments should use past tense. The authors should add a scheme with chemical structures N100-g-GAP and PTPET elastomer referring to Scheme 1 in the manuscript; The 1H NMR of N100-g-GAP, N100, and GAP should be added for characterization; Line 137-138: “ There is a weak peak at about 250 °C for the two samples that were caused by the decomposition of azide groups. “ Is there any experimental evidence or literature to support the decomposition of azide? The stress-strain curves of PTPET elastomers and propellants should be added into the manuscript; How many tests of each sample were done for stress-strain curve? The strain of PTPET elastomer prepared by N100-g-GAP (99.85%) was 43% of strain prepared from GAP(228.45%) in Fig. 5. However, the strain of Propellant from N100-g-GAP (22.96%) was 87% of strain from GAP(26.40%) in Fig 8. What caused this significant difference in strain? The authors should discuss this difference. Line 254-256: “It is obviously displayed that propellants with GAP has a distinctly poor adhesion of RDX &AP particle and binder matrix. There are no apparent inhomogeneities in the distribution of the PTPET (N100-g-GAP) elastomer between RDX and AP.” The SEM images in this manuscript cannot determine the interface of polymer and inorganic particles. The authors should offer higher resolution of SEM images to support their claims.

Reviewer 2 Report

This paper mainly explains about synthesizing a new curing agent N100-g-GAP and using this for elastomer PTPET. The properties of PTPET was analyzed by TGA, DMA, FTIR and mechanical testing. This paper gives the comprehensive study of this grafted material and explains it effect on the elastomer in detail. It was good to understand this new method and its effect. 

These are some of the minor comments that the author should look at before publishing: 

Line 40: "There is a problems" ....the article does not match with the plural verb. It should be "There is a problem" 

Line 57: Define what "carbamates" is so that all readers can understand

Line 60: Define/explain in brief what "click chemistry" is so that reader can understand

In last paragraph of Introduction please explain in brief what is the reason for grafting. What was the problem that this method will solve and help polymer/elastomer community?

Line 142: Why is any temperature above 400C mentioned as suitable for use in practice? Why is this particular number the limit? Explain it.

After this minor revisions, this paper is good to be published 

Round 2

Reviewer 1 Report

The authors have addressed my concerns. Recommend accepting in Polymers.